# TBKIN: Threshold-based explicit selection for enhanced cross-modal semantic alignments

**Zihan Guo**[1], **Xiang Shen**[2,3]*, **Chongqing Chen**[2]

1 Department of Computer Science, Changzhi University, Changzhi, Shanxi, China, 2 College of Information Engineering, Shanghai Maritime University, Pudong, Shanghai, China, 3 School of Computer Science, The University of Sydney, Sydney, New South Wales, Australia

* shenxiang1107@163.com

**Data availability statement:** All relevant datasets are publicly accessible from the VQA database (https://visualqa/index.html) and the REC database (https://cocodataset.org/#home).

## Abstract

Vision-language models aim to seamlessly integrate visual and linguistic information for multi-modal tasks, demanding precise semantic alignments between image-text pairs while minimizing the influence of irrelevant data. While existing methods leverage intra-modal and cross-modal knowledge to enhance alignments, they often fall short in sufficiently reducing interference, which ultimately constrains model performance. To address this gap, we propose a novel vision-language model, the threshold-based knowledge integration network (TBKIN), designed to effectively capture intra-modal and cross-modal knowledge while systematically mitigating the impact of extraneous information. TBKIN employs unified scene graph structures and advanced masking strategies to strengthen semantic alignments and introduces a fine-tuning strategy based on threshold selection to eliminate noise. Comprehensive experimental evaluations demonstrate the efficacy of TBKIN, with our best model achieving state-of-the-art accuracy of 73.90% on the VQA 2.0 dataset and 84.60% on the RefCOCO dataset. Attention visualization and detailed result analysis further validate the robustness of TBKIN in tackling vision-language tasks. The model's ability to reduce interference while enhancing semantic alignments underscores its potential for advancing multi-modal learning. Extensive experiments across four widely-used benchmark datasets confirm its superior performance on two typical vision-language tasks, offering a practical and effective solution for real-world applications.

## Introduction

Vision-language tasks represent a rapidly advancing frontier in artificial intelligence research, bridging the domains of computer vision and natural language processing. This interdisciplinary field aims to empower machines with the capability to process, comprehend, and establish meaningful connections between images and text, thereby enhancing performance across a range of downstream tasks such as visual question answering [1,2] and referring expression comprehension [3–5]. State-of-the-art methods [6–8] typically involve two key

**Funding:** This study was funded by a grant from Shanxi Provincial Education Department (2024L351, Zihan Guo), and a grant from China Scholarship Council (202408310293, Xiang Shen). This study was also supported by Changzhi University Key Laboratory of Intelligent Human-Machine Cooperative Control (Zihan Guo). The funders had no role in study design, data collection and analysis, decision to publish, or preparation of the manuscript.

**Competing interests:** The authors have declared that no competing interests exist.

stages: pretraining models on large-scale image-text datasets to capture foundational knowledge, followed by fine-tuning these models for specific downstream tasks through transfer learning techniques. Pretraining strategies predominantly encompass contrastive learning [9], generative pretraining [10], and alignment pretraining [11], while transfer learning approaches include prompt tuning [12], feature adaptation [13], and other specialized methods. These advancements underscore the potential of vision-language models to tackle complex real-world applications by enabling machines to interpret multi-modal data with human-like proficiency.

Modeling fine-grained semantic alignments between images and text across heterogeneous modalities represents a pivotal aspect of enhancing the performance of vision-language models. The core challenge lies in mapping input images and text into a unified latent space. Early endeavors addressed this challenge by independently pre-training separate models to extract visual and textual features. Nonetheless, such approaches often fall short in establishing robust mechanisms for effectively aligning multi-modal information. To bridge this gap, Jiang et al. [14] introduced a cross-modal implicit relational reasoning and alignment framework, which not only achieves global alignment of visual and textual embeddings but also mitigates the issue of intra-modal information distortion. A notable hurdle in this domain is the substantial cost associated with manually annotating dense semantic alignments between images and text. Yet, deep learning models necessitate extensive datasets for effective pre-training. Consequently, most contemporary methodologies implicitly acquire cross-modal semantic alignments through weakly supervised learning paradigms. Specifically, they leverage the transformer architecture [15] as the backbone, modeling fine-grained semantic alignments based on coarse-grained image-text alignment supervision.

Moreover, image region-word pairs correspond to local cross-modal matches, which may exhibit unreliability or inconsistency from a global perspective, thereby leading to the modeling of inaccurate correlations. To address this limitation, the cross-modal confidence-aware network [16] further evaluates the confidence of local semantic alignments by analyzing the interplay between local and global semantic alignments. This approach enables the establishment of more precise and robust fine-grained cross-modal local semantic alignments. During the same period, Jia et al. [17] introduced ALIGN, a large-scale model that leverages noisy text supervision to learn fine-grained semantic correspondences, demonstrating robust performance in image-text retrieval and zero-shot classification. The aforementioned methods significantly improve the efficiency and accuracy of fine-grained semantic alignments between image-text pairs through diverse technical approaches. Furthermore, their empirical results underscore the substantial potential of fine-grained semantic alignments in advancing vision-language tasks.

The vision-language models outlined above prioritize effective and precise fine-grained semantic alignments but frequently falter in filtering out irrelevant information during their deployment in downstream tasks. This oversight often culminates in subpar performance, as the intrusion of distracting data can impede the models' ability to accurately interpret and align cross-modal information. As illustrated in Fig 1, consider a visual question answering scenario where an image depicts a bathroom equipped with various fixtures, and the question posed is "What is on the toilet?" Traditional fine-tuning strategy employed by existing models often exhibit limitations in accurately isolating relevant objects from the visual context. Specifically, they are prone to erroneously associating spatially adjacent but irrelevant objects with the query—for instance, misidentifying "toilet paper" mounted on the wall as being on the toilet. Furthermore, they can be distracted by distant objects within the scene, leading to incorrect interpretations. The principal challenges facing these models include a. dealing with irrelevant information, where the models struggle to discern and disregard objects that,

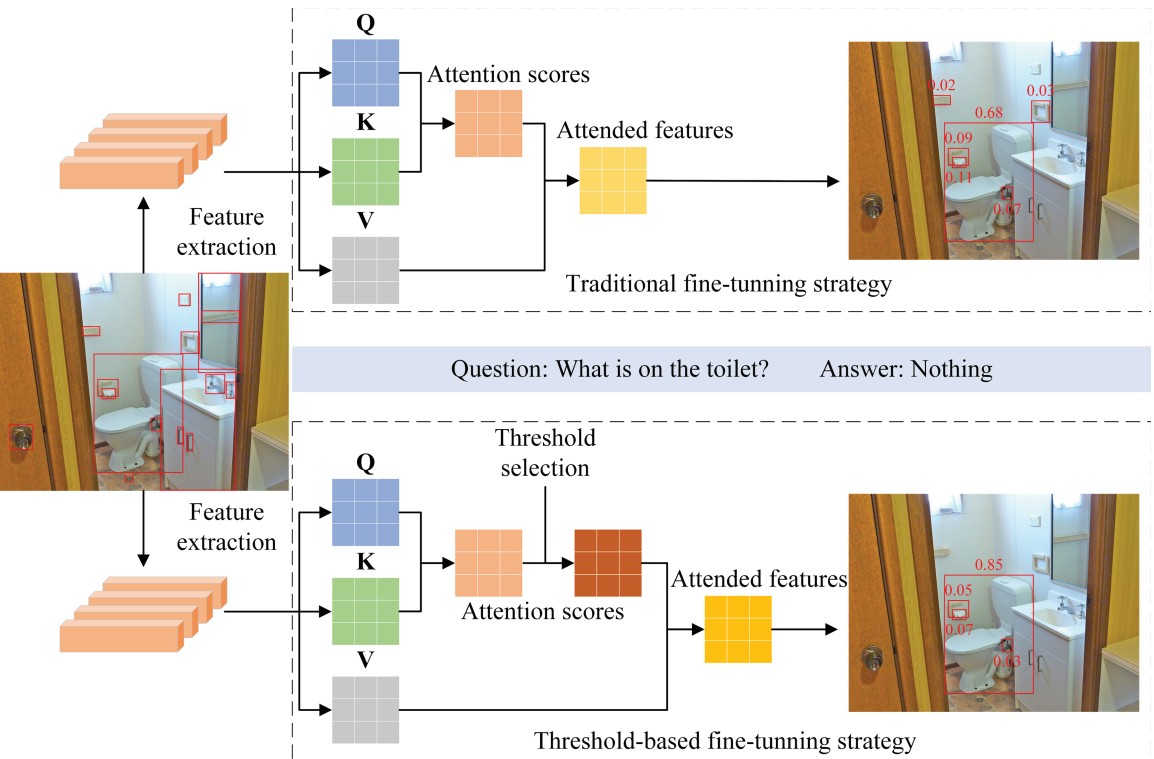

**Fig 1. Illustration of the traditional fine-tuning strategy and the proposed threshold-based fine-tuning strategy.** In comparison to the traditional fine-tuning strategy, the threshold-based fine-tuning strategy incorporates an explicit threshold selection mechanism to sparsify attention scores, thereby enabling the model to capture more effective and targeted feature representations.

while present in the image, hold no relevance to the query at hand, and b. resisting distraction from background details, where models are often misled by visually prominent elements in the image that, despite their salience, contribute nothing to the answer. These inadequacies precipitate inaccurate answers and diminished robustness, particularly in real-world scenarios where images are laden with complex or noisy details. Thus, there arises an imperative need for a solution that not only cements semantic alignments but also curtails the influence of irrelevant information to bolster the robustness and accuracy of vision-language models.

In response to this need, our paper introduces the threshold-based knowledge integration network (TBKIN), a novel vision-language model that augments the foundations laid by ROSITA [18] with pivotal innovations. TBKIN models the intra-modal knowledge within a given image-text pair by leveraging a pre-trained object detector and a scene graph parser. It calculates semantic similarities between image regions and text words by predicting region labels for the image regions and employing a pre-trained word embedding model to capture the cross-modal knowledge. After constructing a unified scene graph of the image-text pair through this process, TBKIN extracts the knowledge entries embedded within it via anchor objects. A pre-trained feature extractor and word embedding layers are then utilized to derive the image and text representations from these knowledge entries, which are subsequently concatenated to form a labeled fusion representation. The fusion representation is then fed into a transformer architecture, which comprises a multi-head attention mechanism and a point-wise feedforward layer. During pre-training, TBKIN adopts a multi-task learning objective that includes knowledge masking tasks based on three distinct masking strategies, alongside

an image-text matching task, to enhance cross-modal semantic alignments. Following the pre-training phase, TBKIN integrates a fine-tuning strategy based on threshold selection to systematically mitigate irrelevant and noisy elements from the input data. As shown in Fig 1, the model fine-tuned with the threshold-based strategy outperforms the model using the traditional fine-tuning strategy in terms of visual feature representation, exhibiting fewer and less significant irrelevant visual objects with respect to the task of generating correct answers. This strategy enhances the model's ability to focus on salient features, thereby significantly optimizing its performance in downstream tasks. Extensive experimental studies and analyses demonstrate the effectiveness of TBKIN in completing vision-language tasks, showcasing its ability to outperform existing methods on multiple benchmarks. The contributions of this work are outlined as follows:

1. We propose TBKIN, a novel vision-language model that augments cross-modal semantic alignments while curtailing the influence of irrelevant information.
2. We introduce a threshold selection-based fine-tuning strategy for downstream tasks that mitigates the impact of irrelevant and distracting information in input data, further optimizing the model's performance.
3. TBKIN showcases robust performance on two typical vision-language tasks across four widely-used benchmark datasets, affirming its effectiveness and potential for real-world applications.

## Related work

We present a concise review of research within the vision-language domain, with particular emphasis on both unimodal and multi-modal models.

### Unimodal models

Unimodal models are primarily designed to process single modalities, such as images or text. For instance, ViT [19] processes input images by dividing them into small patches, which are then linearly projected and encoded through multiple layers, enabling the application of transformer architectures to tasks like image classification and segmentation. Experimental results demonstrate that directly employing a pure transformer on sequences of image patches can effectively achieve vision tasks while requiring fewer computational resources during training. Shi introduced TransNeXt [20], a novel vision backbone that integrates aggregated attention for efficient information mixing and convolutional GLU for robust local modeling, achieving state-of-the-art performance across various model sizes. In the domain of natural language processing (NLP), numerous unimodal models continue to evolve, enabling the handling of more complex tasks with improved performance and generalization. Unlike supervised computer vision models, NLP models learn language representations through generative tasks, eliminating the need for annotated data. The advent of transformer architecture has significantly transformed the sequential modeling paradigm, which previously relied heavily on RNNs and LSTMs. For example, BERT models [21] leverage transformer architecture for pre-training on large-scale text corpora to capture intricate linguistic patterns. OpenAI's GPT, currently the most advanced large language model, also adopts transformer architecture. GPT employs an auto-regressive approach to generate text by iteratively predicting the next word until a complete sentence is formed, exhibiting exceptional performance across diverse NLP tasks.

## Multi-modal models

While unimodal models exhibit robust performance in processing single-modality data, the evolution of multi-modal learning has spurred extensive research into integrating these models to tackle more complex multi-modal challenges. Unlike unimodal models, which are trained on data from a single modality, vision-language multi-modal models leverage large-scale paired image-text corpora, such as image captioning datasets, for pretraining. For instance, TextHarmony [22] employs a hybrid architecture combining ViT, MLLM, and diffusion models to achieve comprehensive visual text understanding and generation. TextHarmony addresses the inherent modality gap between vision and language during training through Slide-LoRA technology, fostering a more coherent generation process. Similarly, BLIP-2 [23] introduces a lightweight querying transformer (Q-Former) to enable efficient cross-modal alignment and information extraction between a frozen image encoder and a frozen large language model (LLM). During the pretraining phase, Q-Former is fine-tuned to transform visual features extracted by the image encoder into visual prompts for LLM, significantly enhancing the model's capability in vision-to-language generation tasks. Additionally, vision-language models integrate diverse techniques, including generative pretraining, feature adaptation, knowledge distillation [24], and multilingual enhancement [25], to further optimize their performance.

## Methodology

This section presents the overall framework and specific details of TBKIN. The overall framework of TBKIN is illustrated in Fig 2. The TBKIN framework consists of four key components. First, in the knowledge extraction phase, TBKIN constructs a unified scene graph based on the input image and question, and subsequently extracts knowledge entities from this graph. Next, in the feature representation phase, TBKIN derives image features and text features from the extracted knowledge entities. These features are then concatenated using two distinctive labels to form a fused representation. Following this, in the pre-training phase, TBKIN employs three distinct masking strategies to optimize the model. Finally, during the fine-tuning phase, TBKIN utilizes a threshold-based selection strategy to eliminate interference from irrelevant information, thereby enhancing task performance.

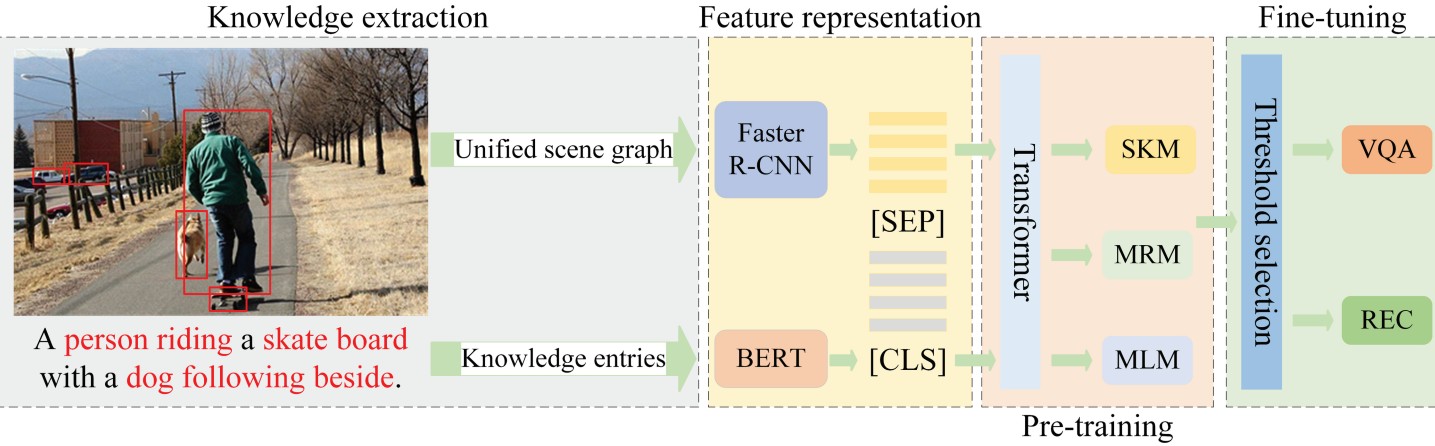

**Fig 2. The overall framework of TBKIN.**

## Knowledge extraction

Initially, we construct unified scene graphs from the input image-text pairs to encapsulate both intra-modal and cross-modal knowledge representations. Subsequently, anchor objects are identified within these scene graphs, and relevant knowledge entries are systematically extracted. The detailed process of knowledge extraction is visually depicted in Fig 3. In the following sections, we will provide a comprehensive and detailed explanation of this process.

**Unified scene graph construction.** Given an image-text pair, we employ a unified scene graph $G = <V, E, S>$ to represent the intra-modal and cross-modal knowledge contained therein [26]. Among them, the vertex set $V$ denotes the image regions present in the image and the words contained in the text, the edge set $E$ signifies the pairwise relationships between the vertices, and the similarity set $S$ encompasses the similarities between the vertices, i.e., the weights of the edges. Specifically, for the given image, we consider the image regions extracted by a pretrained object detector as the vertices in $V$, and the Intersection over Union (IoU) score [27–29] between each pair of image regions as their similarities in $S$, and only the image region pairs with IoU scores greater than zero have edges in $E$. On the other hand, we utilize a scene graph parser [30] to encode the information contained in the given text, encompassing keywords, attributes, and relationships of objects. The keywords are regarded as the vertices in $V$, while the relationships (i.e., object-attribute or object-relation) correspond to the edges in $E$. The similarities in $S$ are determined by the co-occurrence frequencies of the object-attribute pairs and the object-relation pairs across the entire dataset. To address potential discrepancies in similarity distributions across modalities, we independently normalize the similarity scores for each modality. This approach enables effective modeling of intra-modal knowledge within the scene graph.

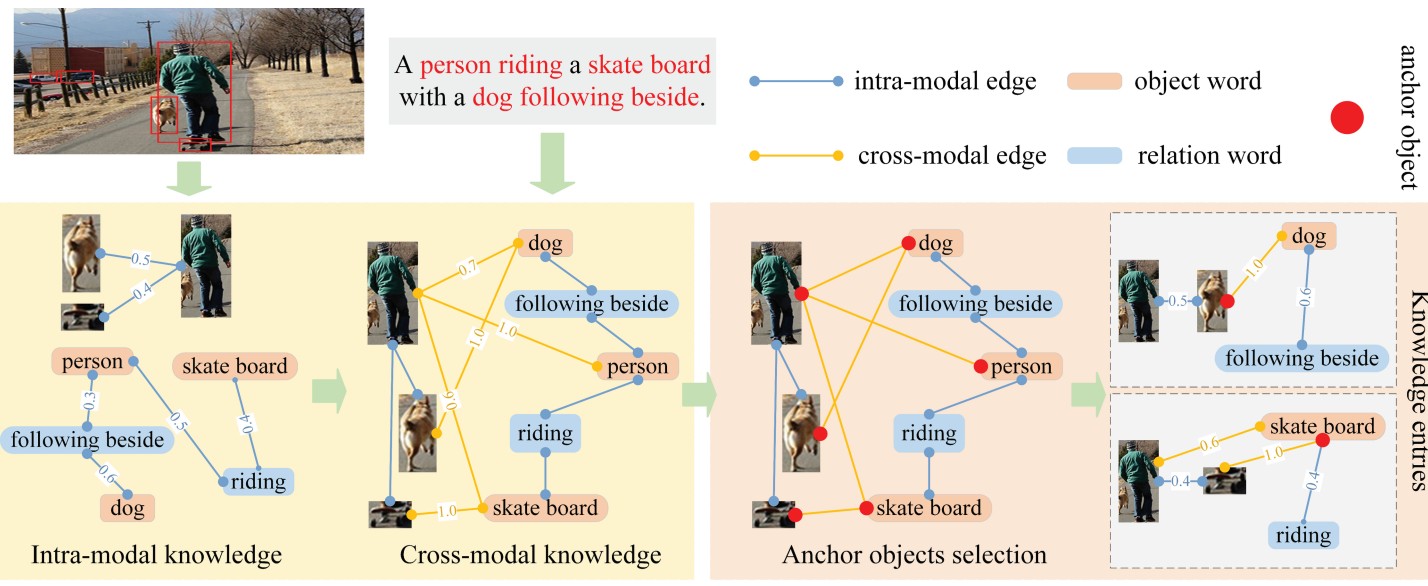

**Fig 3. The process of knowledge extraction.** In the construction of the scene graph, the image regions from the visual input and the words from the textual input form the vertices. The intra-modal and cross-modal relationships between these vertices are represented by the weights of the edges. During the extraction of knowledge entries, we start from anchor objects, i.e., vertices that possess at least one cross-modal edge, and extract the corresponding knowledge entries associated with these anchors.

To model cross-modal knowledge, we establish pseudo-semantic alignments between image regions and textual keywords. Specifically, we predict region labels for the image regions and compute their semantic similarities to the textual keywords using a pretrained word embedding model [31]. These semantic similarities are used to align the image regions with relevant keywords. The resulting relationships between image regions and textual keywords form the cross-modal edges in $\mathbf{E}$, with their similarity scores serving as the corresponding edge weights. This unified framework facilitates a robust representation of both intra-modal and cross-modal relationships in the scene graph.

**Extraction of knowledge entries.** Having constructed the unified scene graph, we proceed to extract knowledge entries, each of which corresponds to an anchor object. The selection of anchor objects is defined by the following criteria: an anchor object must be a vertex (either an image region or a textual word) that is connected by at least one cross-modal edge within the scene graph. This ensures that attribute words and relation words, which lack direct connections to image regions, are excluded from the set of anchor objects. Such a selection mechanism guarantees that the extracted knowledge entries are grounded in cross-modal relationships, thereby enhancing their relevance and utility. The corresponding knowledge entry $g(v) \subseteq \mathbf{G}$ of an anchor object $v \in \mathbf{V}$ is obtained by merging three subgraphs of $\mathbf{G}$ as follows:

$$g(v) = \mathbf{G}_{\text{intra}}(v) \cup \mathbf{G}_{\text{cross}}(v) \cup \mathbf{G}_{\text{intra}}(\mathbf{G}_{\text{cross}}(v)) \tag{1}$$

where $\mathbf{G}_{\text{intra}}(v)$ represents the relationships between $v$ and its directly connected vertices of the same modality, $\mathbf{G}_{\text{cross}}(v)$ represents the relationships between $v$ and its directly connected cross-modal vertices, and $\mathbf{G}_{\text{intra}}(\mathbf{G}_{\text{cross}}(v))$ represents the relationships between the cross-modal vertices directly connected to $v$ and their directly connected vertices of the same modality.

## Feature representations

**Image representations.** Similar to many other vision-language models, we use the pretrained faster R-CNN model [32] to extract regional features from the input images and utilize them as the image representations. To ensure the quality of these representations, we only select $m$ image regions with the highest detection probabilities. The $a$-th image region is described by a visual feature $u_a \in \mathbb{R}^{2048}$ and a positional feature $p_a \in \mathbb{R}^5$ [33]. Then, we employ two linear projections to integrate the two features into a single $d$-dimensional image representation $x_a \in \mathbb{R}^d$ as follows:

$$x_a = \mathbf{W}_{\text{u}}^{\text{T}} u_a + \mathbf{W}_{\text{p}}^{\text{T}} p_a \tag{2}$$

where $\mathbf{W}_{\text{u}} \in \mathbb{R}^{2048 \times d}$ and $\mathbf{W}_{\text{p}} \in \mathbb{R}^{5 \times d}$ are projection matrices. Finally, we obtain the image representation $\mathbf{X} \in \mathbb{R}^{m \times d}$.

**Text representations.** We employ a methodology similar to the one described in [21] for processing the input text. First, we segment the input text into individual words and then adjust it to contain $n$ words through cropping or filling. Next, we utilize two separate word embedding layers to project each word $w_i$ and its corresponding index $i$ denoting its absolute position in the text into two vectors. These vectors are then combined to derive the text representation $y_i \in \mathbb{R}^d$ as follows:

$$y_i = \text{wordembed}(w_i) + \text{indexembed}(i) \tag{3}$$

Finally, we obtain the text representation $\mathbf{Y} \in \mathbb{R}^{n \times d}$.

**Fusion representations.** Once the representations of the input image and the input text are obtained, they are concatenated and two special labels [SEP] and [CLS] are inserted to obtain the fusion representation $\mathbf{Z}$ as follows:

$$\mathbf{Z} = [x_1, x_2, ..., x_m, [\text{SEP}], y_1, y_2, ..., y_n, [\text{CLS}]] \qquad (4)$$

where the [SEP] label is utilized to separate the image representation from the text representation, while the [CLS] label is used to predict whether the input image and the input text are matched or not. Finally, we fed the fusion representation $\mathbf{Z}$ into an $L$-layer transformer, where each layer of the transformer is composed of a multi-head attention mechanism and a pointwise feedforward layer [21]. To facilitate optimization, the transformer also incorporates residual connection [34] and layer normalization [35] technologies.

## Masking strategies and learning objective

Most vision-language models rely on masking strategies to accomplish pretraining tasks. In TBKIN, we introduce a structural knowledge masking (SKM) strategy [18], which leverages explicit probabilities to enhance alignment learning. Additionally, we incorporate two masking strategies (MRM [6] and MLM [21]), both based on random probabilities, to ensure comprehensive modeling of all input information. This multi-faceted approach enables robust representation learning by balancing explicit structural knowledge with stochastic modeling techniques.

Given a knowledge entry, SKM first masks the anchor object and subsequently masks the intra-modal and cross-modal information based on explicitly calculated masking probabilities. Specifically, higher masking probabilities are assigned to intra-modal information that exhibits greater similarity to the anchor object. This design aims to prevent the model from exploiting prior knowledge encapsulated within the information, thereby mitigating potential interference (e.g., the frequent association of the word "grass" with the attribute "green"). Conversely, lower masking probabilities are applied to cross-modal information with higher semantic similarity to the anchor object. This strategy encourages the model to reconstruct the masked anchor object by leveraging semantically relevant information from the complementary modality, implicitly enhancing cross-modal semantic alignments. In contrast to SKM, MRM and MLM randomly mask input image regions or textual words, respectively, to complement SKM by promoting a holistic understanding of the input semantics. Following the methodologies outlined in [6] and [18], we jointly adopt classification-based and regression-based loss functions to optimize the model. Detailed operational procedures of these masking strategies are beyond the scope of this paper. Readers are referred to [6] and [18] for comprehensive insights.

Following the approaches described in [6] and [8], we adopt a multi-task learning objective that combines knowledge masking tasks based on the three aforementioned strategies and an image-text matching task to facilitate model pretraining. The knowledge masking tasks involve training the model to reconstruct masked tokens by leveraging contextual information, achieved by selectively masking portions of the input data. This process is designed to enhance the model's ability to comprehend and integrate visual and linguistic information. Furthermore, the image-text matching task aims to train the model to associate image content with corresponding textual descriptions, thereby fostering the discovery and establishment of cross-modal semantic consistency between the input image and text. Collectively, these tasks contribute to a robust pretraining framework that aligns visual and textual representations while promoting a deeper understanding of multi-modal interactions.

### Fine-tuning strategy based on threshold selection

Following the pretraining phase, we adopt a fine-tuning strategy based on threshold selection to eliminate irrelevant and interfering information within the input data, thereby enhancing the model's performance on downstream tasks, specifically visual question answering and referring expression comprehension. To achieve this, we refine the multi-head attention mechanism within the transformer architecture. The conventional multi-head attention mechanism [15] operates by taking queries and key-value pairs as inputs and generating attended features as outputs, which are essentially weighted sums of the values. The computation of these weights involves a compatibility function that processes the queries and their corresponding keys. Specifically, the mechanism computes the dot product between the queries $\mathbf{Q}$ and the keys $\mathbf{K}$, subsequently normalizing these values by dividing them by $\sqrt{k}$ to obtain attention scores $\mathbf{H}$. These scores are then passed through a softmax function to derive the probabilistic weights $\mathbf{W}$ assigned to each value. Finally, the attended features $\mathbf{F}$ are computed as the weighted sum of the values $\mathbf{V}$. The mathematical formulation of this process is expressed as follows:

$$\mathbf{H} = \frac{\mathbf{Q}\mathbf{K}^{\mathrm{T}}}{\sqrt{k}} \tag{5}$$

$$\mathbf{W} = \mathrm{softmax}(\mathbf{H}) \tag{6}$$

$$\mathbf{F} = \mathbf{W}\mathbf{V} \tag{7}$$

where $\mathbf{Q}$, $\mathbf{K}$, and $\mathbf{V}$ represent the sets of vectors consisting of queries, keys, and values respectively, and $k$ is the dimension of $\mathbf{K}$.

The traditional scaled dot-product attention mechanism, as described above, is effective in amplifying salient regional image features and critical textual keywords. However, it inherently lacks the ability to suppress irrelevant or noisy information contained within the input data, which can adversely affect the model's performance on downstream tasks. To mitigate this limitation, we propose a threshold-based explicit selection strategy that selectively focuses the model's attention on the most task-relevant inputs. Our approach involves introducing a threshold $T$ to filter the attention weight matrix $\mathbf{W}$. Specifically, for each attention weight in the matrix, if its value is below the predefined threshold $T$, the corresponding position in the attention scores $\mathbf{H}$ is set to $-\infty$. This modification ensures that, after reapplying the softmax function, the weights corresponding to these positions converge to 0, thereby effectively suppressing the influence of irrelevant information. The mathematical formulation of this process is outlined as follows:

$$\mathbf{H}'_{ij} = \begin{cases} \mathbf{H}_{ij} & \mathbf{W}_{ij} \geq T \\ -\infty & \mathbf{W}_{ij} < T \end{cases} \tag{8}$$

$$\mathbf{W}' = \mathrm{softmax}(\mathbf{H}') \tag{9}$$

$$\mathbf{F} = \mathbf{W}'\mathbf{V} \tag{10}$$

where $\mathbf{H}'$ is the new attention scores, and $\mathbf{W}'$ is the new weight matrix. Fig 4 shows the calculation process of the threshold-based scaled dot-product attention, where $q$ is the dimension of $\mathbf{Q}$. It is important to emphasize that this strategy is intentionally omitted during the pre-training phase to enable the model to achieve more comprehensive intra-modal and cross-modal semantic alignments.

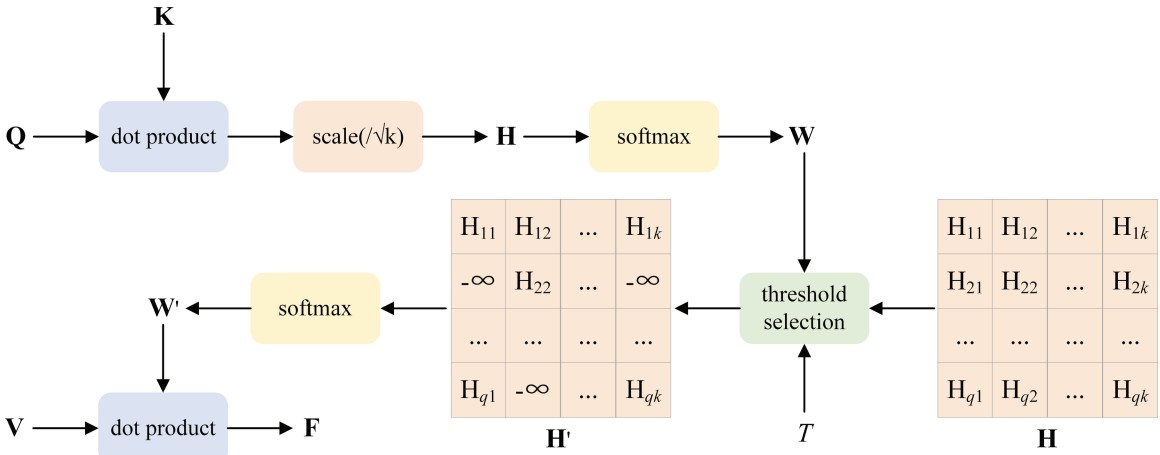

**Fig 4. The calculation process of the threshold-based scaled dot-product attention.**

## Experiments and results

For the pretraining of TBKIN, we utilize a large-scale dataset comprising image-text pairs, which serves as the foundational training corpus to initialize the model. Following pretraining, we rigorously evaluate the performance of TBKIN on two benchmark vision-language tasks, leveraging four widely recognized datasets to ensure comprehensive and robust assessment. Additionally, to examine the efficacy of TBKIN in mitigating the impact of distracting information and enhancing task-relevant attention, we conduct an in-depth analysis through attention visualization techniques. This visualization not only provides qualitative insights into the model's internal mechanism but also validates its ability to selectively focus on salient features while disregarding irrelevant or noisy inputs.

## Pretraining of the model

Similar to [6], we construct the pretraining dataset based on four datasets, namely SBU Captions [36], COCO Captions [37], Visual Genome Captions [38], and Conceptual Captions [39]. The pretraining dataset contains 9.5M training instances and 155K validation instances, respectively.

When calculating the similarities between image region labels and text words, we set a minimum confidence threshold of 0.5 in order to achieve a balance between precision and recall. Following [21], for the input image and text, we utilize the Faster R-CNN model to extract a fixed number of 36 image regional features and adopt the BPE strategy to tokenize the text into a maximum of 50 words. TBKIN's backbone is a 12-layer transformer, which includes 768 hidden units and 12 attention heads. TBKIN's parameters are initialized using the publicly available BERT-base model [21]. This initialization ensures that the model starts from a robust semantic representation, which is crucial for effective downstream performance. Following initialization, the model is trained for up to 40 epochs with a batch size of 512 to optimize its performance. In the MRM and MLM tasks, we set the masking probabilities to 15%. For detailed parameter settings related to the structural knowledge masking strategy, please refer to [18].

## Downstream tasks

Upon the completion of the pretraining phase, we proceed to fine-tune TBKIN on two downstream vision-language tasks, leveraging task-specific datasets to evaluate its performance comprehensively. This fine-tuning process ensures that the pretrained model adapts effectively to the unique demands of each task, thereby enabling a thorough assessment of its capabilities in handling real-world vision-language challenges.

**Visual question answering (VQA).** The VQA task takes images and free-form, open-ended text questions related to the image content as input, requiring the model to generate accurate natural language answers. We utilize the widely recognized benchmark dataset VQA 2.0 [40], which is divided into three sets: train (83K images and 444K questions), validation (41K images and 214K questions), and test (81K images and 448K questions). We fed the [CLS] label of the fusion representation into a linear classifier to predict the correct answer from a vocabulary of 3129 possible answers.

**Referring expression comprehension (REC).** The objective of the REC task is to accurately localize the target object in an image based on a reference expression expressed in natural language. To evaluate the model's performance, we adopt three REC datasets derived from COCO images [41]: RefCOCO, RefCOCO+, and RefCOCOg. Both RefCOCO and Ref-COCO+ are divided into four subsets: train (120K queries), validation (11K queries), testA (6K queries related to people), and testB (6K queries related to objects). RefCOCOg is divided into three subsets: train (81K queries), validation (5K queries), and test (10K queries). The representation for each image region is used to predict a ranking score and a refined bounding box.

## Experimental results

To comprehensively evaluate the effectiveness of our proposed TBKIN model, we conduct a comparative analysis against several state-of-the-art vision-language models. The experimental results, as presented in Table 1, highlight TBKIN's superior performance, with the best results for each column denoted in bold. The findings demonstrate that TBKIN achieves consistent and robust performance across all evaluated downstream tasks, underscoring the efficacy of its integrated intra-modal and cross-modal knowledge modeling framework. Furthermore, the fine-tuning strategy incorporating threshold-based selection is validated as a critical contributor to the model's enhanced accuracy and task adaptability. These results collectively affirm the technical advancements and practical utility of TBKIN in addressing vision-language challenges. Among them, LXMERT [42] consists of three encoders, i.e., the object relationship encoder, the language encoder, and the cross-modal encoder, to learn the alignments and relationships between visual concepts and language semantics. UNITER [6] achieves outstanding pre-training results through the utilization of conditional masking, which involves masking regions and languages based on a comprehensive observation of images and text. Additionally, the WRA technique is employed to promote fine-grained alignments across different modalities during pretraining. VILLA [7] is a robust vision-language model that utilizes the free adversarial training strategy and incorporates KL-divergence-based regularization. The re-attention framework [2] dynamically modifies the visual attention maps based on the information contained in the answers and autonomously controls the contribution of re-attention to model training using a gate mechanism. ERNIE-ViL [33] is a knowledge-enhanced method for acquiring integrated vision-language representations through the formulation of scene graph prediction tasks during the pre-training stage. ROSITA [18], similar to our proposed approach, improves semantic alignments between

**Table 1. Comparisons with state-of-the-art models on VQA and REC.**

| Model | VQA 2.0 | RefCOCO | | | RefCOCO+ | | | RefCOCOg | |
|---|---|---|---|---|---|---|---|---|---|
| | test-dev | val | testA | testB | val | testA | testB | val | test |
| LXMERT [42] | 72.42 | – | – | – | – | – | – | – | – |
| UNITER [6] | 72.70 | 81.24 | 86.48 | 73.94 | 75.31 | 81.30 | 65.68 | 74.31 | 74.51 |
| VILLA [7] | 73.59 | 81.65 | 87.40 | 74.48 | 76.05 | 81.65 | 65.70 | 75.90 | 75.93 |
| Re-attention [2] | 71.60 | – | – | – | – | – | – | – | – |
| ERNIE-ViL [33] | 72.62 | – | – | – | 74.02 | 80.33 | 64.74 | – | – |
| ROSITA [18] | 73.40 | 84.40 | 87.77 | 78.77 | 76.06 | 82.01 | 67.40 | 77.41 | **78.24** |
| LOIS [1] | 72.78 | – | – | – | – | – | – | – | – |
| LGR [3] | – | 83.69 | 86.42 | **79.25** | 73.50 | 78.36 | 65.02 | 74.14 | 74.23 |
| TBKIN(ours) | **73.90** | **84.60** | **87.98** | 78.36 | **76.72** | **82.55** | **68.00** | **78.31** | 78.17 |

cross-modal knowledge through the use of unified scene graphs and masking strategies. However, it does not further mitigate the interference of irrelevant information. The more refined VQA model LOIS [1], which does not rely on bounding boxes, integrates a mutual relation attention module to effectively capture intricate visual semantic relationships between background information and instance objects, and enhances the model's capacity for visual reasoning by focusing on important word-related questions. LGR NET [3] for REC tasks introduces a prediction token to effectively model cross-modal features and extends the details of textual features such as spatial information to sufficiently utilize the textual features.

The aforementioned models represent state-of-the-art advancements in vision-language modeling, achieving remarkable progress in pre-training strategies and cross-modal alignments. However, a fundamental limitation shared by these models is their inability to explicitly address the challenge of filtering out irrelevant information during downstream tasks. This oversight often results in suboptimal performance in noisy or complex scenarios, where the capability to discern and disregard distracting data is crucial for ensuring both accuracy and robustness. Consequently, the persistent interference of irrelevant information hampers their effectiveness in real-world applications, particularly those involving cluttered visuals and intricate textual inputs, thereby limiting their overall utility. In contrast to these models, TBKIN introduces a threshold-based knowledge integration network that not only enhances cross-modal alignments but also actively mitigates the influence of irrelevant information. By employing a sparse selection method during fine-tuning, TBKIN effectively filters out distracting data, ensuring robust performance even in noisy scenarios. As demonstrated by our experimental results, TBKIN consistently showcases robust performance across multiple benchmarks.

## Ablation studies

We perform ablation experiments on four benchmark datasets for the VQA and REC tasks to investigate the impact of our explicit threshold-based selection strategy and varying thresholds on the performance of vision-language models in downstream tasks. The experimental results are detailed in Tables 2–5 and Fig 5, with the best-performing results in each column highlighted in bold. Here, Base refers to the ROSITA model without any sparse attention mechanism. The numbers following TBKIN denote the specific values of the threshold $T$, where a larger $T$ indicates that more attention weights with smaller values are filtered out. In other words, a higher $T$ excludes more information deemed less relevant to the model's ability to complete the target task. The results show that TBKIN achieves superior performance across all four datasets for both downstream tasks when an appropriate threshold is selected,

**Table 2. Experimental results of TBKIN on the test-dev set of VQA 2.0.**

| Model | Yes/No | Number | Other | Overall |
|---|---|---|---|---|
| Base | 89.69 | 55.29 | 63.58 | 73.40 |
| Local-4 | 76.02 | 40.71 | 48.62 | 59.01 |
| Local-14 | 81.97 | 46.52 | 54.03 | 64.69 |
| Local-7 | 78.83 | 42.42 | 49.88 | 60.96 |
| Local-8 | 76.78 | 41.95 | 49.11 | 59.69 |
| TBKIN-0.000001 | 89.57 | 55.55 | 63.73 | 73.45 |
| TBKIN-0.000003 | 89.68 | 56.01 | 64.31 | 73.82 |
| TBKIN-0.0000038 | 89.77 | **56.18** | **64.37** | **73.90** |
| TBKIN-0.000004 | **89.81** | 55.80 | 64.32 | 73.85 |

**Table 3. Experimental results of TBKIN on RefCOCO.**

| Model | val | testA | testB |
|---|---|---|---|
| Base | 84.40 | 87.77 | **78.77** |
| TBKIN-0.000000008 | 84.30 | 87.72 | 78.57 |
| TBKIN-0.000000009 | 84.49 | **88.09** | 78.57 |
| TBKIN-0.0000000095 | **84.60** | 87.98 | 78.36 |
| TBKIN-0.00000001 | 84.57 | 87.96 | 78.06 |

**Table 4. Experimental results of TBKIN on RefCOCO+.**

| Model | val | testA | testB |
|---|---|---|---|
| Base | 76.06 | 82.01 | 67.40 |
| TBKIN-0.000000004 | 76.35 | 82.15 | **68.02** |
| TBKIN-0.0000000045 | 76.13 | **82.62** | 67.65 |
| TBKIN-0.00000000475 | **76.72** | 82.55 | 68.00 |
| TBKIN-0.000000005 | 76.68 | 82.59 | 67.63 |

**Table 5. Experimental results of TBKIN on RefCOCOg.**

| Model | val | test |
|---|---|---|
| Base | 77.41 | **78.24** |
| TBKIN-0.00009 | 77.68 | 77.67 |
| TBKIN-0.0001 | 77.72 | **78.24** |
| TBKIN-0.0001001 | **78.31** | 78.17 |
| TBKIN-0.000101 | 77.74 | 77.98 |

thereby validating its effectiveness and interpretability. Additionally, we explore integrating ROSITA with other sparse attention methods, such as local attention, for the VQA task. In local attention, each query is constrained to attend only to its neighboring nodes. However, as demonstrated by the experimental results (rows 2-5 in Table 2, where the numbers following Local represent the size of the attention window), the performance of local attention is significantly inferior to that of TBKIN, which utilizes a threshold-based filter. This further underscores the robustness and efficiency of our TBKIN approach.

## Attention visualization

To further substantiate the advantages of our TBKIN model, we present a comparative analysis of visual attention maps between the Base model and TBKIN, as illustrated in Fig 6. This

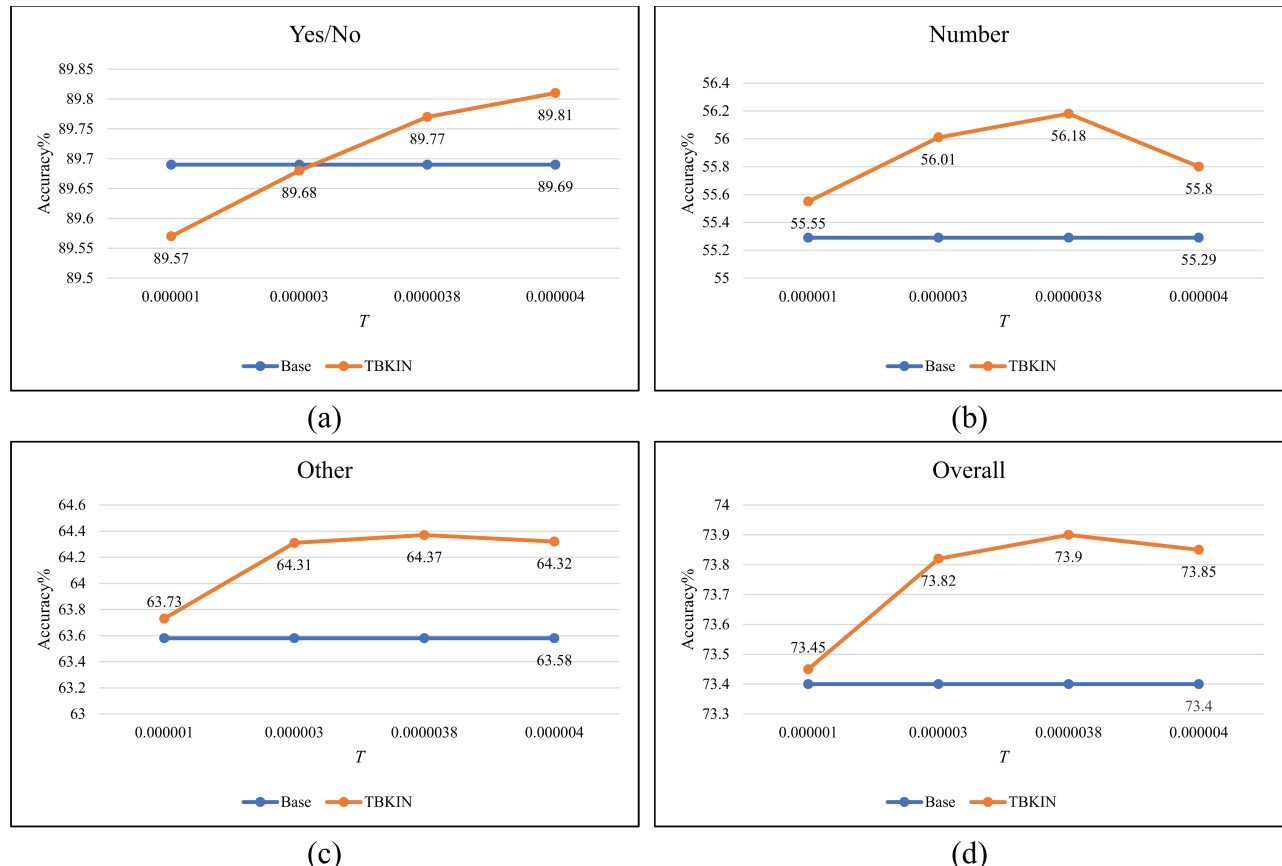

**Fig 5. Experimental results of TBKIN and Base on the test-dev set of VQA 2.0.** (**a**) The accuracy of Yes/No based on different thresholds. (**b**) The accuracy of Number based on different thresholds. (**c**) The accuracy of Other based on different thresholds. (**d**) The overall accuracy based on different thresholds.

visualization is conducted on the VQA task to highlight the enhanced focus and precision of TBKIN. For the first input instance, the Base model misallocates attention to the ears of zebras, which are unrelated to the facial regions central to the question. In contrast, TBKIN demonstrates remarkable precision by exclusively concentrating on the two visible zebra faces, aligning perfectly with the task requirements. In the second instance, while the Base model correctly focuses on the two elephants, its attention dispersion extends to irrelevant elements such as river rocks and branches. Conversely, TBKIN exhibits superior selectivity by almost exclusively attending to the two elephants, thereby ensuring a more accurate response to the input question. For the final instance, both models successfully focus on the toilet. However, a closer examination reveals that TBKIN's attention is notably more precise and compact, which is hypothesized to contribute to improved answer prediction accuracy by minimizing distractions from extraneous visual elements. These findings collectively underscore the effectiveness of TBKIN in refining attention mechanisms, enabling it to focus on task-relevant regions with greater precision and thereby enhancing its performance on vision-language tasks.

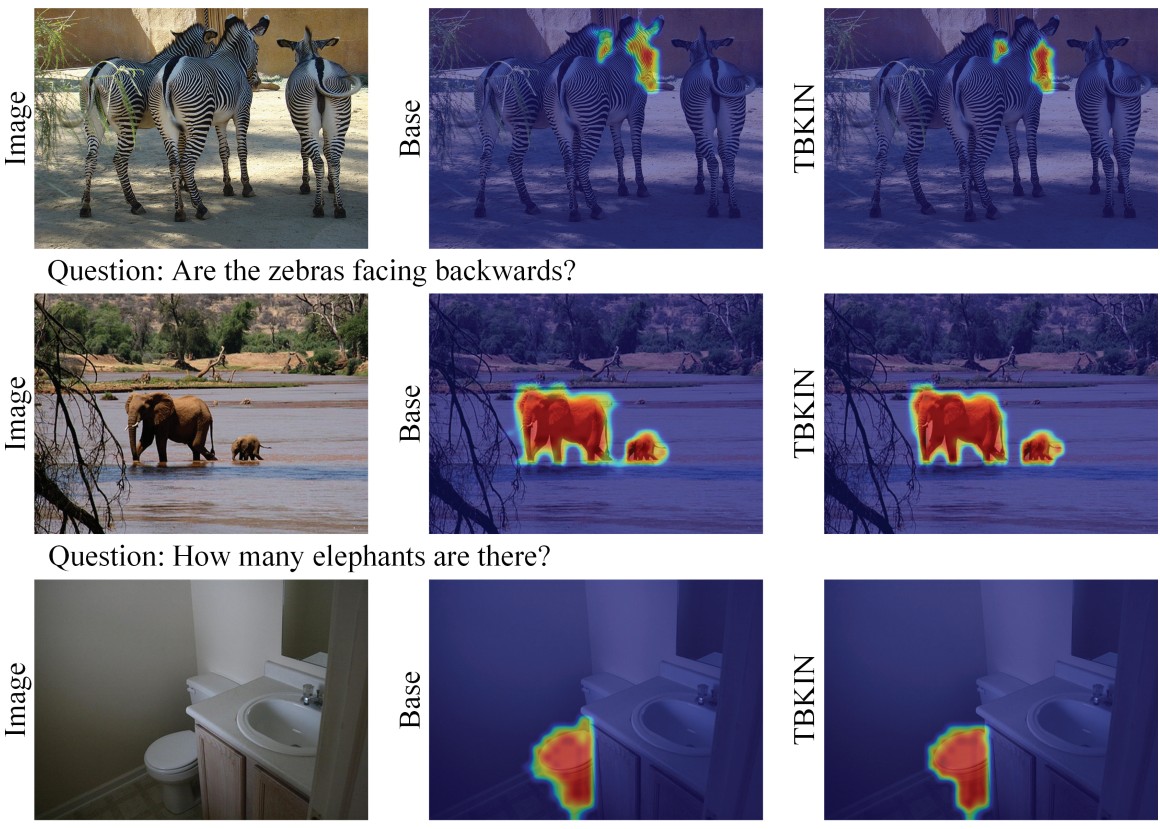

**Fig 6. The attention visualization results of the Base model and TBKIN.** The first column displays the input images and corresponding questions. The second column illustrates the visual attention maps generated by the Base model based on the input instances, while the third column presents the visual attention maps produced by our TBKIN model for the same inputs.

## Conclusion

In this work, we propose a novel vision-language model, termed TBKIN (threshold-based knowledge integration network), which introduces a unified scene graph framework to simultaneously learn intra-modal and cross-modal knowledge during the pre-training phase. By leveraging diverse masking strategies, TBKIN enhances the semantic alignment between visual and textual representations, enabling robust cross-modal understanding. Additionally, for downstream tasks, TBKIN incorporates a threshold-based sparse attention mechanism, which effectively mitigates the influence of irrelevant information, thereby further optimizing model performance. Extensive experiments conducted on four datasets across two vision-and-language tasks, complemented by ablation studies and attention visualization analyses, collectively validate the efficacy and interpretability of TBKIN. The results highlight its ability to achieve state-of-the-art performance while maintaining transparency in decision-making processes. We envision that the methodologies and insights presented in this study will not only advance the field of vision-language research but also inspire innovative solutions in other multi-modal domains. The integration of unified scene graphs and threshold-based mechanisms provides a promising direction for future exploration in complex multi-modal learning scenarios.

## Author contributions

**Conceptualization:** Zihan Guo.

**Data curation:** Zihan Guo.

**Formal analysis:** Zihan Guo.

**Funding acquisition:** Zihan Guo, Xiang Shen.

**Investigation:** Zihan Guo.

**Methodology:** Zihan Guo.

**Project administration:** Zihan Guo.

**Resources:** Zihan Guo, Xiang Shen, Chongqing Chen.

**Software:** Zihan Guo, Xiang Shen, Chongqing Chen.

**Supervision:** Zihan Guo.

**Validation:** Zihan Guo.

**Visualization:** Zihan Guo, Chongqing Chen.

**Writing – original draft:** Zihan Guo.

**Writing – review & editing:** Zihan Guo.

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
