## [Decision Letter · Decision Letter 0]

PONE-D-24-60420Threshold-based knowledge integration vision-language modelPLOS ONE

Dear Dr. Guo,

Thank you for submitting your manuscript to PLOS ONE. After careful consideration, we feel that it has merit but does not fully meet PLOS ONE’s publication criteria as it currently stands. Therefore, we invite you to submit a revised version of the manuscript that addresses the points raised during the review process.

We look forward to receiving your revised manuscript.

Kind regards,

Yongjie Li

Academic Editor

PLOS ONE

2. In the online submission form, you indicated that your data is available only on request from a third party. Please note that your Data Availability Statement is currently missing [the name of the third party contact or institution / contact details for the third party, such as an email address or a link to where data requests can be made]. Please update your statement with the missing information.

3. Please note that your Data Availability Statement is currently missing [the repository name and/or the DOI/accession number of each dataset OR a direct link to access each database]. If your manuscript is accepted for publication, you will be asked to provide these details on a very short timeline. We therefore suggest that you provide this information now, though we will not hold up the peer review process if you are unable.

Additional Editor Comments:

Among the comments from the reviewers, the authors are expected to particularly clarify the the novelty and the contribution of this work, and clearly differentiate the proposed approach from existing models.

Reviewers' comments:

Reviewer's Responses to Questions

**Comments to the Author**

1. Is the manuscript technically sound, and do the data support the conclusions?

Reviewer #1: Yes

Reviewer #2: Partly

2. Has the statistical analysis been performed appropriately and rigorously? 

Reviewer #1: I Don't Know

Reviewer #2: No

3. Have the authors made all data underlying the findings in their manuscript fully available?

Reviewer #1: Yes

Reviewer #2: No

4. Is the manuscript presented in an intelligible fashion and written in standard English?

Reviewer #1: Yes

Reviewer #2: No

5. Review Comments to the Author

Reviewer #1: Title - Threshold-based knowledge integration vision-language model

In this paper, the authors have proposed a novel vision language model called threshold-based knowledge integration network (TBKIN), which captures intramodal and cross-modal knowledge while mitigating the impact of irrelevant information to improve performance.

Queries and Suggestions (Comments)

1. The title of this paper is not informative enough, please revised

2. Abstract is not well written (IMRaD is a standard format for writing abstracts in an orderly manner ....such as i. Introduction, ii Methodology, iii Results and iv Discussions). A summary of the result is missing from this abstract. Please rewrite the abstract.

3. The introduction can be enriched by briefly describe the state-of-the-art in the title of this study and provide more recently related references to support foundation of this studies to better context for the current research. I will recommend citing recently published related papers;

a. Developing Nigeria Multilingual Languages Speech Datasets for Antenatal Orientation. In H. Florez, & H. (. Astudillo (Ed.), Applied Informatics. ICAI 2024. Communications in Computer and Information Science. 2237, pp. 157-170. Springer, Cham. doi:https://doi.org/10.1007/978-3-031-75147-9_11

b. EASESUM: an online abstractive and extractive text summarizer using deep learning technique. IAES International Journal of Artificial Intelligence (IJ-AI), 13(2), 1888-1899. Retrieved from https://ijai.iaescore.com/index.php/IJAI/article/view/23221

4. The problem that author is trying to address in this paper is not clear yet, (clearer motivation for study is required)

5. In 4-5- lines, authors should summarize the literature gaps identified before starting the methodology (Kindly relate your work in the context of existing work, what are the shortcomings of existing studies?)

6. Author should re-check the correctness of equations especially equations 3 to 8

7. Can you further explain how your work succeed beyond REF 5, 6, and 15.

8. Services of language expert are required (be mindful of tenses)

9. Image quality needs to be improved

10. Ensure that all Figures, Tables and Equations are properly labeled and referenced.

Reviewer #2: The clarity of the paper should be improved. It often is hard to understand, and the text leaves it unclear, what the intentions are. The introduction leaves it unclear TBKIN learning strategy on intra-modal and cross-modal knowledge based on unified scene graphs and masking strategies.

Please clarify: Is the VLM only used during training or during rollout? For the method without VLM, in which stages has it been left out?

Please explain initialization, as this is barely mentioned in the rest of the main text.

Ablations regarding the VQA and REC policy are not clear. In general, the architecture of the policy is not clear from the paper.

Could you provide some graphical visualization of the table? I believe this would make the claimed more apparent.

The figs are hard to understand.

6. PLOS authors have the option to publish the peer review history of their article (what does this mean?). If published, this will include your full peer review and any attached files.

Reviewer #1: **Yes: **Sunday A. Ajagbe

Reviewer #2: No

---

## [Author Response · Author response to Decision Letter 1]

28 Mar 2025

Manuscript ID: PONE-D-24-60420

Paper Title: Threshold-based knowledge integration vision-language model

Dear Editor,

We sincerely appreciate your letter and the insightful comments provided by the reviewers on our manuscript titled Threshold-based knowledge integration vision-language model (PONE-D-24-60420). The reviewers’ feedback is invaluable, offering both constructive guidance for refining and enhancing our paper and significant insights that will inform our ongoing research. We have meticulously examined each comment and implemented the necessary revisions to address them comprehensively. To uphold transparency and streamline the revision review process, we have systematically annotated all modifications within the manuscript. Deleted content is meticulously marked with strikethroughs, while newly incorporated material addressing journal requirements and editorial feedback is distinctly highlighted in red. Revisions in response to Reviewer #1’s comments are marked in blue, and those addressing Reviewer #2’s comments are clearly identified in orange. This structured approach ensures clarity and facilitates a comprehensive assessment of the revisions made in accordance with the feedback provided.

We extend our heartfelt gratitude to the editor and reviewers for their meticulous and constructive feedback, which has profoundly strengthened the rigor and clarity of our manuscript. Their insightful suggestions have notably improved the theoretical framework, experimental methodology, and the overall reliability of our findings. We are deeply appreciative of the opportunity to resubmit the revised manuscript and eagerly await further guidance to ensure its excellence.

Once again, we sincerely thank you for your invaluable expertise, patience, and support throughout this process.

Best regards,

Zihan Guo, Xiang Shen, Chongqing Chen

Response to journal requirements

1. Please ensure that your manuscript meets PLOS ONE’s style requirements, including those for file naming.

Response: We have meticulously reviewed our manuscript in accordance with the PLOS ONE style templates and implemented the necessary revisions. Specifically, we have:

1. Revised the formatting of tables (Tables 1-5) and adjusted the citation style for multiple tables (line 633).

2. Renamed all figure files to align with the journal’s requirements.

3. Applied the “et al.” convention for references with more than six authors, listing only the first six authors followed by “et al.” (references 6, 12, 15, 18, 19, 22, 27, 33, 37, 38).

2. In the online submission form, you indicated that your data is available only on request from a third party. Please note that your Data Availability Statement is currently missing [the name of the third party contact or institution / contact details for the third party, such as an email address or a link to where data requests can be made]. Please update your statement with the missing information.

Response: We sincerely apologize for the oversight in our initial description of data availability. It was incorrect to state that the data are available only upon request from a third party. In fact, our data are openly accessible and have been deposited in public repositories. We will promptly update the statement to reflect this information accurately and ensure full compliance with journal requirements.

3. Please note that your Data Availability Statement is currently missing [the repository name and/or the DOI/accession number of each dataset OR a direct link to access each database]. If your manuscript is accepted for publication, you will be asked to provide these details on a very short timeline. We therefore suggest that you provide this information now, though we will not hold up the peer review process if you are unable.

Response: Thank you for your reminder. We have now updated our Data Availability Statement as follows:

All relevant datasets are publicly accessible from the VQA database (https://visualqa.org/index.html) and the REC database (https://cocodataset.org/#home).

We confirm that these links provide direct access to the datasets used in our study.

Response to additional editor comments

Additional editor comments:

Among the comments from the reviewers, the authors are expected to particularly clarify the novelty and the contribution of this work, and clearly differentiate the proposed approach from existing models.

Response: We sincerely appreciate your feedback and the opportunity to further clarify the novelty and contribution of our work. In response to the reviewers’ comments, we have carefully elaborated on the unique aspects of our proposed approach and how it distinguishes itself from existing models. We have addressed this in detail in our responses to Reviewer #1’s comments 4, 5, and 7 and Reviewer #2’s comment 1. Specifically, the revisions include, but are not limited to, the following key aspects:

1. Highlighting the innovative aspects of our methodology and its theoretical advancements.

2. Clearly differentiating our approach from prior models by emphasizing its unique features and performance improvements.

3. Providing a more detailed discussion of the practical contributions of our work to the field.

We hope these clarifications adequately address your concerns and demonstrate the distinct novelty and contribution of our study. Should any additional modifications be required, we are happy to address them promptly.

Response to reviewers’ comments

Reviewers’ comments:

Reviewer’s responses to questions

Comments to the author

1. Is the manuscript technically sound, and do the data support the conclusions?

Reviewer #1: Yes

Reviewer #2: Partly

Response: We sincerely appreciate the opportunity to address the concerns regarding the technical soundness and data support for the conclusions in our manuscript. In response to the comments from reviewers, we have carefully revised the manuscript to ensure its rigor, clarity, and validity. Specifically, the revisions include, but are not limited to, the following key aspects:

1. Clarity and intent of the paper: We have improved the overall clarity of the manuscript by refining the language, structure, and presentation to ensure that the research objectives, methodology, and contributions are well-articulated and easy to follow.

2. Initialization method and learning strategy of the TBKIN model: We have provided a more detailed and transparent explanation of the initialization process and learning strategy employed for the proposed TBKIN model to enhance reproducibility and technical rigor.

3. Improved presentation of figures and tables: We have revised the figures and tables to ensure they are more intuitive, clearly labeled, and effectively support the conclusions drawn in the manuscript.

These revisions directly address the concerns raised by the reviewers and strengthen the technical soundness of the manuscript. For detailed responses, please refer to our replies to Reviewer #1’s comment 4 and Reviewer #2’s comments 1-7. We believe the revised manuscript now provides robust data and analysis that fully support the conclusions drawn. Should any additional modifications be required, we are happy to address them promptly.

2. Has the statistical analysis been performed appropriately and rigorously?

Reviewer #1: I Don’t Know

Reviewer #2: No

Response: We deeply appreciate the opportunity to address the concern regarding the appropriateness and rigor of the statistical analysis in our manuscript. In response to the comments from reviewers, we have carefully revised the manuscript to ensure the statistical analysis is appropriately executed and comprehensively described. The revisions include, but are not limited to, the following key aspects:

1. Model details and learning strategy: We have elaborated on the specific details of the proposed model, including its learning strategy and the use of the VLM model, to enhance the clarity and reproducibility of our approach.

2. Clearer description and analysis of experiments and results: We have improved the description of the model’s initialization method, presented experimental results in a more graphical and intuitive manner, and provided a detailed comparison with state-of-the-art models. These revisions ensure a robust and transparent analysis of our findings.

For more detailed responses, please refer to our replies to Reviewer #1’s comment 7 and Reviewer #2’s comments 2-6. We believe these revisions have strengthened the statistical rigor and overall quality of the manuscript. Should any additional modifications be required, we are happy to address them promptly.

3. Have the authors made all data underlying the findings in their manuscript fully available?

Reviewer #1: Yes

Reviewer #2: No

Response: Thank you for highlighting the importance of data availability in our manuscript. We greatly appreciate the opportunity to address this concern and ensure full compliance with the PLOS Data policy. In our study, we utilized the VQA 2.0 dataset and three REC datasets derived from COCO images, all of which are publicly available and stored in well-established repositories. To address Reviewer #2’s concern and further enhance transparency, we have updated the Data Availability Statement to explicitly detail how these datasets can be accessed without restriction. Specifically, the relevant data are openly available through the following sources:

VQA 2.0 Dataset: Accessible at https://visualqa.org/index.html.

REC Datasets (COCO-based): Accessible at https://cocodataset.org/#home.

We sincerely apologize for any oversight in our initial submission regarding the clarity of data accessibility. We are committed to upholding the highest standards of data sharing and transparency in our research. We believe these revisions fully address the concerns raised by Reviewer #2 and align our manuscript with the requirements of the PLOS Data policy. Should any additional modifications be required, we are happy to address them promptly.

4. Is the manuscript presented in an intelligible fashion and written in standard English?

Reviewer #1: Yes

Reviewer #2: No

Response: We sincerely thank the reviewers and the editor for their valuable feedback regarding the presentation and language of our manuscript. In response to the comments from reviewers, we have thoroughly revised the manuscript to ensure it is presented in an intelligible fashion and written in clear, standard English. The revisions include, but are not limited to, the following key improvements:

1. Enhanced title: The manuscript title has been revised to be more comprehensive and reflective of the study’s scope.

2. Structured and complete abstract: The Abstract has been restructured to provide a clearer and more complete summary of the research.

3. Enriched introduction: The Introduction section has been expanded to provide a more thorough background and context for the study.

4. Clearer research motivation: The motivation behind the research has been articulated more explicitly to better highlight its significance.

5. Explanation of advantages over existing methods: We have provided a more detailed explanation of how our approach outperforms existing methods.

6. Improved clarity and readability: The overall clarity and readability of the manuscript have been enhanced to ensure it is accessible to a broad audience.

For detailed responses to the specific comments, please refer to our replies to Reviewer #1’s comments 1-4, 7, and 8, as well as Reviewer #2’s comments 1-7. We believe these revisions have significantly improved the presentation and language quality of the manuscript. Should any additional modifications be required, we are happy to address them promptly.

5. Review comments to the author

Reviewer #1:

1. The title of this paper is not informative enough, please revised.

Response: We sincerely thank you for your valuable feedback regarding the title of our paper. In response to your observation that the title was not sufficiently informative, we have carefully revised it to better emphasize the core contributions and focus of our work. The updated title is now:

“TBKIN: Threshold-based explicit selection for enhanced cross-modal semantic alignments”

We believe this version provides clearer and more precise insights into the key contributions and scope of our research. Should any additional modifications be required, we are happy to address them promptly.

2. Abstract is not well written (IMRaD is a standard format for writing abstracts in an orderly manner .... Such as i. Introduction, ii Methodology, iii Results and iv Discussions). A summary of the result is missing from this abstract. Please rewrite the abstract.

Response: We sincerely thank you for your valuable feedback regarding the Abstract. In response to your observation that the Abstract was not well-structured and lacked a summary of results, we have thoroughly revised it in alignment with the IMRaD format. The updated Abstract is provided below for your review:

We believe the revised Abstract now offers a more structured, informative, and complete overview of our research. Should any additional modifications be required, we are happy to address them promptly.

3. The introduction can be enriched by briefly describe the state-of-the-art in the title of this study and provide more recently related references to support foundation of this studies to better context for the current research. I will recommend citing recently published related papers.

Response: We sincerely thank you for your valuable feedback and insightful suggestions. In response to your recommendation, we have enriched the Introduction by providing a concise description of a recent advancement in the field related to our research (lines 71-74). Additionally, we have incorporated your suggested references, as well as another relevant recent publication, to strengthen the foundation of our work. These additions provide a comprehensive background and highlight how our study aligns with current state-of-the-art developments in the field. Should any additional modifications be required, we are happy to address them promptly.

4. The problem that author is trying to address in this paper is not clear yet, (clearer motivation for study is required).

Response: We sincerely thank you for your valuable feedback. We acknowledge that the motivation for our study was not sufficiently clear in the original manuscript. To address this, we have thoroughly revised the Introduction to explicitly state the problem we aim to solve and the significance of our research. Specifically:

1. Clear problem statement: We added a concise description of the research problem in the Introduction, emphasizing its relevance and the gaps in existing approaches (lines 83-102).

2. Enhanced motivation: We expanded the discussio

---

## [Decision Letter · Decision Letter 1]

TBKIN: Threshold-based explicit selection for enhanced cross-modal semantic alignments

PONE-D-24-60420R1

Dear Dr. Guo,

We’re pleased to inform you that your manuscript has been judged scientifically suitable for publication and will be formally accepted for publication once it meets all outstanding technical requirements.

Kind regards,

Yongjie Li

Academic Editor

PLOS ONE

Additional Editor Comments (optional):

Reviewers' comments:

Reviewer's Responses to Questions

**Comments to the Author**

1. If the authors have adequately addressed your comments raised in a previous round of review and you feel that this manuscript is now acceptable for publication, you may indicate that here to bypass the “Comments to the Author” section, enter your conflict of interest statement in the “Confidential to Editor” section, and submit your "Accept" recommendation.

Reviewer #1: All comments have been addressed

Reviewer #2: All comments have been addressed

2. Is the manuscript technically sound, and do the data support the conclusions?

Reviewer #1: Yes

Reviewer #2: Yes

3. Has the statistical analysis been performed appropriately and rigorously? 

Reviewer #1: Yes

Reviewer #2: Yes

4. Have the authors made all data underlying the findings in their manuscript fully available?

Reviewer #1: Yes

Reviewer #2: Yes

5. Is the manuscript presented in an intelligible fashion and written in standard English?

Reviewer #1: Yes

Reviewer #2: Yes

6. Review Comments to the Author

Reviewer #1: The manuscript has been revised and it is more suitable for publication to the best of my knowledge

Reviewer #2: Now manuscript titled "TBKIN: Threshold-based explicit selection for enhanced cross-modal semantic

alignments" is clear. Authors have examined each comment and implemented all the necessary corrections.

7. PLOS authors have the option to publish the peer review history of their article (what does this mean?). If published, this will include your full peer review and any attached files.

Reviewer #1: **Yes: **Sunday Adeola Ajagbe

Reviewer #2: **Yes: **Dr. Biraja Ghoshal

---

## [Editor Report · Acceptance letter]

PONE-D-24-60420R1

PLOS ONE

Dear Dr. Guo,

I'm pleased to inform you that your manuscript has been deemed suitable for publication in PLOS ONE. Congratulations! Your manuscript is now being handed over to our production team.

Kind regards,

on behalf of

Professor Yongjie Li

Academic Editor

PLOS ONE